# Estimation of country-level incidence of early-onset invasive Group B Streptococcus disease in infants using Bayesian methods

**Bronner P. Gonçalves**[1,2]\*, **Simon R. Procter**[1,2], **Sam Clifford**[1],
**Artemis Koukounari**[1,2], **Proma Paul**[1,2], **Alexandra Lewin**[3], **Mark Jit**[1☯], **Joy Lawn**[1,2☯]

**1** Department of Infectious Disease Epidemiology, London School of Hygiene & Tropical Medicine, London, United Kingdom, **2** Maternal, Adolescent, Reproductive & Child Health (MARCH) Centre, London School of Hygiene & Tropical Medicine, London, United Kingdom, **3** Department of Medical Statistics, London School of Hygiene & Tropical Medicine, London, United Kingdom

☯ These authors contributed equally to this work.
\* bronner.goncalves@lshtm.ac.uk

**Data Availability Statement:** All relevant data are within the manuscript, its Supporting Information files and at https://github.com/bronnerpg/GBS.

## Abstract

Neonatal invasive disease caused by Group B Streptococcus (GBS) is responsible for much acute mortality and long-term morbidity. To guide development of better prevention strategies, including maternal vaccines that protect neonates against GBS, it is necessary to estimate the burden of this condition globally and in different regions. Here, we present a Bayesian model that estimates country-specific invasive GBS (iGBS) disease incidence in children aged 0 to 6 days. The model combines different types of epidemiological data, each of which has its own limitations: GBS colonization prevalence in pregnant women, risk of iGBS disease in children born to GBS-colonized mothers and direct estimates of iGBS disease incidence where available. In our analysis, we present country-specific maternal GBS colonization prevalence after adjustment for GBS detection assay used in epidemiological studies. We then integrate these results with other epidemiological data and estimate country-level incidence of iGBS disease including in countries with no studies that directly estimate incidence. We are able to simultaneously estimate two key epidemiological quantities: the country-specific incidence of early-onset iGBS disease, and the risk of iGBS disease in babies born to GBS-colonized women. Overall, we believe our method will contribute to a more comprehensive quantification of the global burden of this disease, inform cost-effectiveness assessments of potential maternal GBS vaccines and identify key areas where data are necessary.

## Author summary

Invasive disease caused by Group B Streptococcus (GBS) in young infants continues to be a major public health problem in both developed and developing countries. However, data on the incidence of this infection during the first week of life are only available for a small number of countries, which has complicated the quantification of the burden of this

**Funding:** This work was supported by a grant (OPP1180644) from the Bill & Melinda Gates Foundation (https://www.gatesfoundation.org/) to the London School of Hygiene & Tropical Medicine (PI: JL). MJ was supported by the National Institute of Health Research (NIHR) Health Protection Research Unit (HPRU) in Modelling and Health Economics, a partnership between Public Health England, Imperial College London, and LSHTM (grant code NIHR200908); and in Immunisation, a partnership between Public Health England and LSHTM (grant reference code NIHR200929). The funders had no role in study design, data collection and analysis, decision to publish, or preparation of the manuscript.

**Competing interests:** The authors have declared that no competing interests exist.

disease globally. In this paper, we develop a Bayesian framework to estimate the incidence of invasive GBS infection that combines data from multiple types of epidemiological studies, with adjustment for relevant factors such as diagnostic methods used in the studies. We present estimates from a series of models, and our results highlight the potential weaknesses of different types of studies and the importance to consider the entire evidence when estimating global burden of invasive neonatal infections. We believe this model is a step toward better quantification of the number of cases in different regions.

## Introduction

Infection by Group B Streptococcus (GBS), a gram-positive bacterium that colonizes the genital and gastrointestinal tracts [1,2], causes morbidity and mortality in babies in the first months of life. With severe clinical manifestations that include sepsis and meningitis, invasive GBS disease in infants aged a week or less is associated with case-fatality risks ranging from 5 to 27% [3]. However, despite its public health importance, until recently only a few studies had tried to assess the global burden of this medical condition [4,5].

In the most comprehensive multi-country analysis to date, Seale and colleagues combined data from different types of studies to estimate the global number of invasive GBS (iGBS) disease cases in babies [5]. In that analysis, knowledge on the pathogenesis of iGBS disease was used to identify studies that provide relevant information: for example, colonization of the genito-urinary tract of mothers by GBS is believed to be necessary, via a mechanism of *in utero* infection or infection at delivery [6], for the development of iGBS disease in neonates aged a week or less [7]. In addition to data on the percentages of mothers colonized by GBS, information on the risk of early-onset, i.e. first 7 days of life, iGBS disease in infants born to GBS-colonized mothers was also used. Combined, these data allowed the authors to estimate country-specific numbers of early-onset iGBS disease cases. Studies that directly estimated the incidence of early-onset iGBS disease among all live births, i.e. regardless of mothers' GBS colonization statuses, were not used in these calculations but were compared to the estimates: direct estimates of incidence from epidemiological studies were lower compared to estimates based on maternal GBS colonization prevalence and data on risk in babies born to GBS-colonized mothers. It was argued that under-estimation is common in incidence studies [5]. For example, in an early study in the United Kingdom, authors estimated under-reporting of 44% and 21% from paediatricians and microbiologists, respectively, and that the number of cases might have been ~20% higher than reported [8]. Consistent with this concern on case capture, a recent study with prospective enhanced surveillance that captured data on iGBS disease in the first 3 months of life using a variety of approaches found that for 53% of the identified cases microbiological data from reference laboratories were not available [9]. As suggested in a review of iGBS disease incidence studies performed in low and middle-income countries [4], under-estimation could also be related to limited access to health care facilities and the fact that a baby's clinical condition might deteriorate rapidly, leading to fatal outcome before samples are collected. Indeed, in another review [3], the proportion of early-onset iGBS disease cases diagnosed in the first 24 hours of life was higher in high income versus low income countries (74% and 31% respectively).

The use of maternal GBS colonization as predictor of early-onset iGBS disease also has its own issues, including our limited understanding of the natural history of GBS *in utero*, variability in the risk of iGBS disease given maternal GBS colonization, as well as limited data on coverage of intrapartum antibiotic prophylaxis (IAP), which affects this risk. To date, no study

has integrated GBS colonization and direct iGBS disease incidence data, while adjusting for variability in both datasets; this would require an evidence synthesis framework that combines different data sources within a realistic model of iGBS disease natural history.

Quantification of incidence, and associated uncertainty, is the first step to estimate the global burden of iGBS disease, including the burden linked to long-term consequences [10], and is essential for priority setting around investment in preventive interventions such as GBS vaccines [11,12] and roll out of antibiotic prophylaxis [13]. Here, we use Bayesian methods to estimate the number of early-onset iGBS disease cases in countries with data on maternal GBS colonization prevalence. We developed hierarchical models to estimate country-level maternal GBS colonization prevalence and the risk of iGBS disease in babies born to GBS-colonized women. Furthermore, in addition to data on GBS colonization and on risk of early-onset iGBS disease in neonates of mothers with GBS colonization, we use data from incidence studies, i.e. studies that estimate the percentage of newborns that develop early-onset iGBS disease regardless of maternal GBS colonization status. Hence, our model allows to synthesise evidence from all these different sources. In the *Results* section, we first describe the different components of the model separately, and then present results of the full model that combines these different types of data.

## Results

To estimate the number of cases of early-onset iGBS disease, we developed a Bayesian model to combine data from different types of studies: (1) studies that estimated the prevalence of GBS colonization during pregnancy; (2) studies that assessed the risk of early-onset iGBS disease in neonates born to GBS-colonized mothers in settings with different intrapartum antibiotic use; and (3) studies that estimated the incidence of early-onset iGBS disease in neonates regardless of maternal GBS colonization status.

In the following subsections, the different components of the Bayesian model (**Fig 1**) are described, including underlying assumptions and data. In the subsections *GBS colonization during pregnancy* and *Early-onset invasive GBS disease in babies born to GBS-colonized*

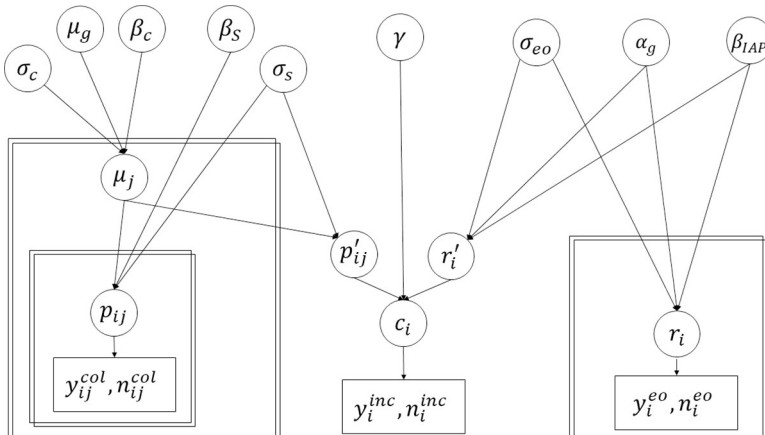

**Fig 1.** The structures of the hierarchical model on maternal GBS colonization (left) and of the hierarchical model on the risk of early-onset iGBS disease in babies born to colonized mothers (right) are shown. The component of the full model that uses data on early-onset iGBS disease incidence as well as functions of parameters estimated in the two first components is presented in the middle. Eqs 1 – 8 in the text describe model parameters and distributional assumptions. Circles represent parameters; rectangles with single border, in the lower part of the figure, correspond to observations and rectangles with double borders represent the hierarchical structure of the data.

*mothers*, hierarchical models are presented as independent analyses; and in the subsection *Synthesis with incidence data*, the models described in the first two subsections are combined with incidence studies.

## GBS colonization during pregnancy

Since the pathogenesis of early-onset iGBS disease involves colonization of the maternal genito-urinary tract by GBS bacteria [2], determining the prevalence of maternal GBS colonization informs the proportion of newborns at risk. In a recent systematic review, data from studies reporting prevalence of GBS colonization during pregnancy were summarized; the methods and inclusion criteria used in the review are described in detail elsewhere [14]. In modelling GBS colonization prevalence, we use studies identified in that review.

Overall, data from 325 studies were included (**Fig 2**); 82 countries had at least one study to inform country-level estimates of maternal GBS colonization prevalence. The number of studies per country ranged from 1 to 31, and the median number of participants in each study was 349 (range 35–17,430). The median prevalence of GBS in these studies was 14.6%.

These studies varied in two aspects that influence sensitivity of GBS detection [7]: the anatomical sites from which samples were collected (vaginal sampling versus recto-vaginal sampling) and the method used for microbiological diagnosis. Regarding the latter, microbiological approaches were categorized in two for this analysis: selective agar (with or without enrichment) and unselective agar (which has lower sensitivity). As shown in **Fig A in S1 Appendix**, these factors influence prevalence estimates. In this analysis, 60% of the studies used the most sensitive diagnostic combination (recto-vaginal sampling and selective agar for culture).

To estimate country-level GBS colonization prevalence in pregnant women, we developed a Bayesian hierarchical model:

$$\mu_j \sim Normal\left(\mu_g + \beta_c X_j^c, \sigma_c^2\right) \tag{1}$$

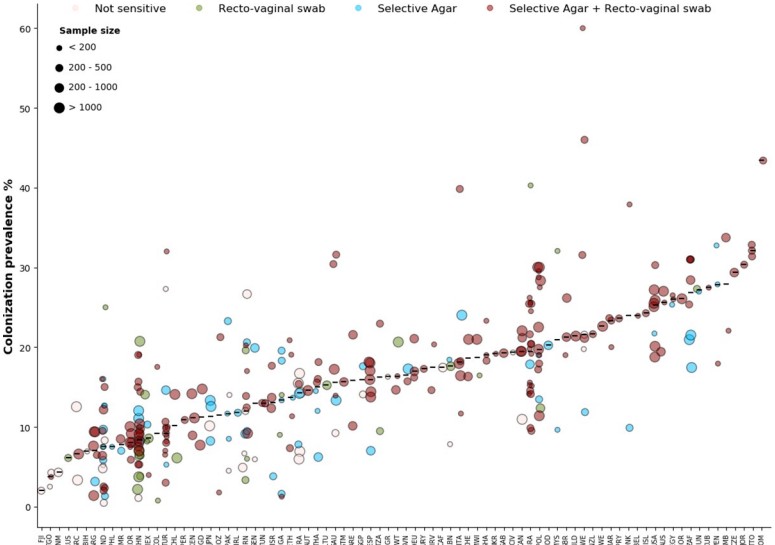

**Fig 2.** Data used in the hierarchical model of maternal GBS colonization. Study-specific percentages of mothers colonized by GBS (circles) are presented (y-axis) by country (ISO3 codes are shown in the x-axis; arranged in order of increasing median prevalence). Colors represent different combinations of microbiological methods and anatomical sampling sites. Horizontal lines represent median maternal GBS colonization prevalence of the studies in each country. The diameter of the circles is determined by the sample size in each study (see upper left corner of the figure).

$$logit \left( p_{ij}^{col} \right) \sim Normal \left( \mu_j + \beta_s X_{ij}^S, \sigma_s^2 \right) \qquad (2)$$

$$y_{ij}^{col} \sim Binomial \left( p_{ij}^{col}, n_{ij}^{col} \right) \qquad (3)$$

where the logit-prevalence in a country $j$, $\mu_j$, was assumed to be dependent on the logit-scale global prevalence, $\mu_g$, and standardized country-level variables, $X_j^c$; $\beta_c$ is a vector of country-level regression coefficients. Between-country variation that is independent of covariates is represented by the standard deviation $\sigma_c$. In this hierarchical model, prevalence $p_{ij}^{col}$ in individual studies was modelled, on the logit scale, as a function of two binary variables (sampling site and microbiological method); the intercept $\mu_j$ was the logit-prevalence in the country where the study was performed. A common within-country between-study variance, $\sigma_s^2$, is assumed. The number of GBS-colonized pregnant women in study $i$ performed in country $j$, $y_{ij}^{col}$, followed a binomial distribution with parameters $n_{ij}^{col}$, the study sample size, and $p_{ij}^{col}$. $\beta_s$ represents the vector of coefficients at the study level; and $X_{ij}^S$, the vector of covariate values for study $i$ in country $j$.

The set of country-level variables used in this analysis was similar to that used in the previous review [5] and selected based on knowledge of the disease process (see *Methods* section for a more detailed discussion on variables). To the covariates in the previous review [5], we added country-level antibiotic coverage during respiratory infections in children, as a surrogate for antibiotic use that could influence GBS carriage. These variables were standardized, i.e. the mean was subtracted, and the resulting value, divided by the standard deviation, before inclusion in the model. The selection of country-level variables included in the final model is described in the *Methods* section. Although Equs 1–3 present the centered parameterization of this model, to minimize divergences observed in the estimation process, a non-centered parameterization was used to sample the posterior distribution (see additional information in the *Methods* section). Regression coefficients were assigned weakly informative normal priors, *Normal ~ (0, 1)* and the parameter $\mu_g$ was assigned the prior *Normal ~ (-1, 1)*, that corresponds to the belief that the prevalence of GBS colonization in pregnant women is below 50%. Standard deviation parameters were assigned uniform priors, *Uniform ~ (0, 5)*. Prior predictive distributions for the models presented in this and the following subsection (**Figs B and C in S1 Appendix**) and sensitivity analyses that used other prior assumptions are shown in the **S1 Appendix**.

Mixed predictive checks, as defined in [15,16] for hierarchical models, are presented in **Fig D in S1 Appendix**. Posterior medians and 95% intervals of study- and country-level coefficients as well as standard deviation parameters are shown in **Table 1**. As expected, vaginal sampling and use of less sensitive microbiological methods were associated with lower prevalence. For the following variables, there was evidence of association with GBS colonization prevalence at the country-level: antibiotics coverage during respiratory infections, prevalence of female obesity, gross national income per capita, neonatal mortality, and HIV prevalence. In **Fig 3**, estimates (posterior medians and 95% intervals) of country-level prevalences are presented for the 82 countries with maternal GBS colonization data.

## Early-onset invasive GBS disease in babies born to GBS-colonized mothers

Only a small percentage of babies born to GBS-colonized mothers develop iGBS disease, and this risk is reduced by the administration of antibiotics during delivery, i.e. intrapartum antibiotic prophylaxis (IAP) [17]. Data to estimate the risk of early-onset iGBS disease come from

**Table 1. Maternal GBS colonization hierarchical model.** Coefficients at the study and country levels and standard deviation parameters. ATB = antibiotics; LRI = lower respiratory tract infections; GNI = gross national income per capita.

| | Median | 95% interval |
|---|---|---|
| **Country-level covariates** | | |
| *Percent coverage of ATB for LRI* | 0.30 | (0.07–0.53) |
| *Maternal education* | -0.18 | (-0.48–0.11) |
| *GNI* | 0.16 | (0.03–0.29) |
| *Neonatal mortality* | 0.22 | (0.00–0.44) |
| *HIV prevalence* | 0.19 | (0.06–0.32) |
| *Obesity prevalence* | 0.20 | (0.08–0.31) |
| **Study-level covariates** | | |
| *Swab site* | -0.23 | (-0.39 - -0.07) |
| *Culture method* | -0.30 | (-0.47 - -0.13) |
| **Standard deviation parameters** | | |
| $\sigma_c$ | 0.33 | (0.23–0.45) |
| $\sigma_s$ | 0.52 | (0.47–0.58) |

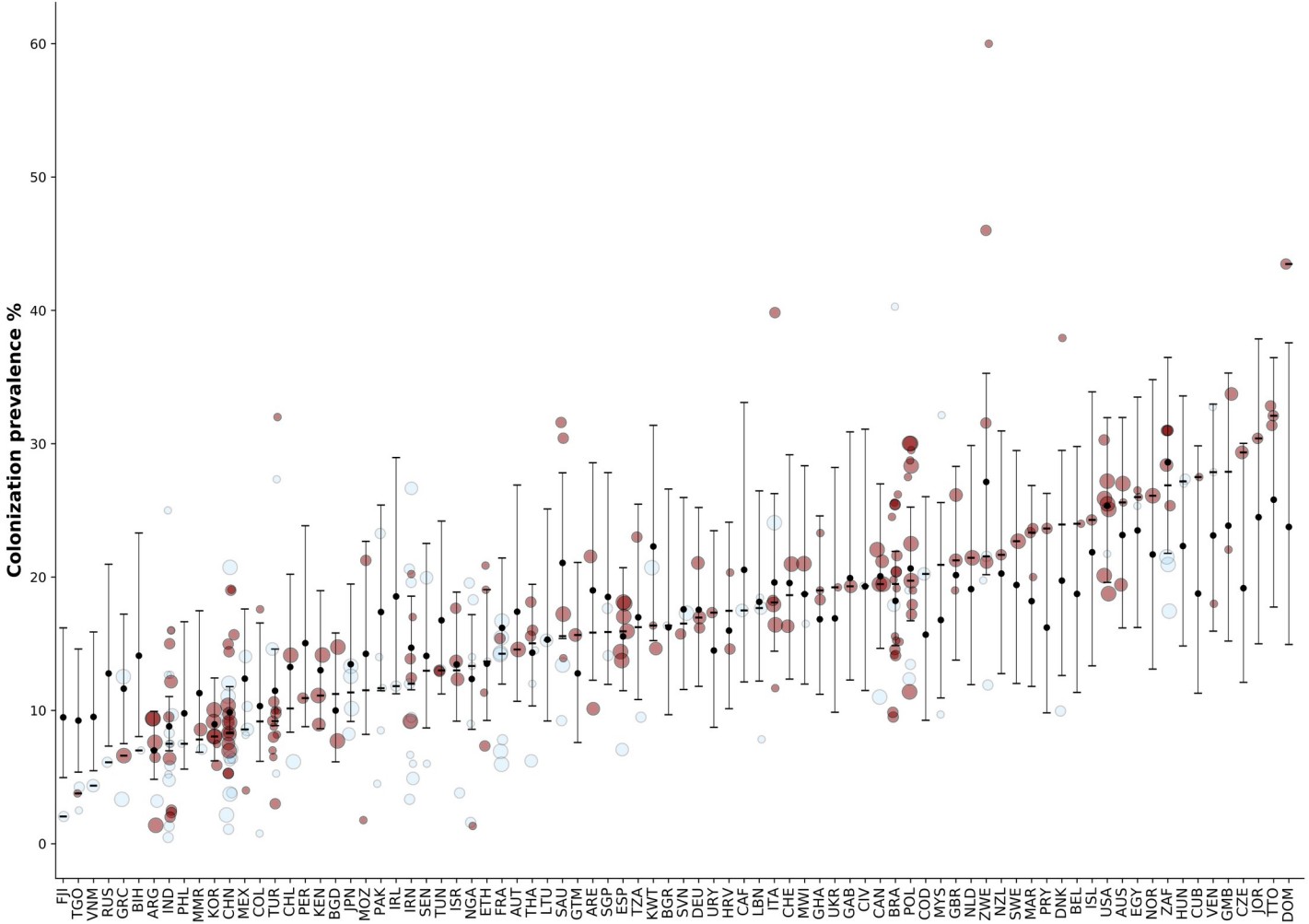

**Fig 3.** Estimated maternal GBS colonization prevalence (y-axis) in countries (x-axis) with epidemiological data. Red circles represent studies that used sensitive GBS detection protocols. Black circles represent posterior medians and lines, 95% intervals; these estimates correspond to country-level prevalences as quantified by sensitive methods, i.e. inverse logit function of $\mu_j$, and not to estimated prevalences in new GBS colonization studies, which would need to incorporate between-study variation and sampling variation.

studies that were described in a review [18]; information on intrapartum antibiotic use was reported in these studies. 9/28 studies were performed in the United States of America, and 14/28, in Europe. Sample sizes ranged from 216 to 3,819; and numbers of early-onset iGBS disease cases ranged from 0 to 24.

We analysed these data with the following hierarchical model:

$$\alpha_i \sim Normal\ (\alpha_g, \sigma_{eo}^2) \tag{4}$$

$$logit\ (r_i) = \alpha_i + \beta_{IAP}X_i^{IAP} \tag{5}$$

$$y_i^{eo} \sim Binomial\ (r_i, n_i^{eo}) \tag{6}$$

where study-specific intercepts, $\alpha_i$, were assumed to be normally distributed with mean $\alpha_g$ and standard deviation $\sigma_{eo}$. The inverse logit of intercept $\alpha_i$ represents the risk in the study population $i$ if IAP were not used; and $\beta_{IAP}$ is the coefficient of the association between study-level antibiotic coverage ($X_i^{IAP}$) and risk of early-onset iGBS disease. $y_i^{eo}$ corresponds to the number of early-onset GBS cases in study $i$ and follows a binomial distribution with parameters $r_i$, the risk of early-onset iGBS disease in study $i$, and $n_i^{eo}$, the sample size of study $i$. The priors for $\alpha_g$, $\sigma_{eo}$ and $\beta_{IAP}$ were *Normal ~ (-4, 1)*, *Uniform ~ (0, 5)* and *Normal ~ (0, 1)*, respectively (see also sensitivity analyses in **S1** *Appendix*); the prior for $\alpha_g$ reflects the expected low risk of iGBS disease ($< 10\%$). This model has lower widely applicable information criterion (WAIC) [19] compared to a model with fixed intercept. We also fit a model with an additional variable that categorized countries in two groups (North America and Europe, 23 studies; and Africa and Asia, 5 studies): no clear association between this variable and risk of iGBS disease was observed; hence we only present results for the model described in Eqs 4–6. Predictive checks for this model are shown in **Fig E in S1 Appendix**.

The posterior medians (95% intervals) of $\alpha_g$ and $\beta_{IAP}$ are -4.12 (-4.92, -3.42) and -0.03 (-0.05, -0.02), respectively. **Fig 4** shows the predicted risk (shaded areas) of early-onset iGBS disease in new studies and the estimated risks in studies included in this analysis. Where IAP is not used, the risk of early-onset iGBS disease in babies born to GBS-colonized mothers is on average 1.6% (inverse logit of $\alpha_g$).

## Synthesis with incidence data

In the analysis by Seale and colleagues [5], the incidence of early-onset iGBS disease was calculated by multiplying the estimated prevalence of GBS colonization in pregnant women and the IAP coverage-adjusted risk of disease in babies born to colonized mothers. Data from studies that directly estimated incidence were not used in the calculation, as it has been argued that these studies under-estimate incidence, for example due to non-optimal access to care and rapid fatality [3]. In this subsection, we combine the models presented in the two previous subsections with data from incidence studies. In our primary analysis, we included only those studies considered to be less biased (N = 10 studies) in a recent review [3]; in this subsection we refer to this analysis as full model I. We also performed a secondary analysis that included all incidence studies with relevant data (N = 30 studies) described in the same review; we refer to this analysis as full model II. More information about these studies is presented in the *Methods* section.

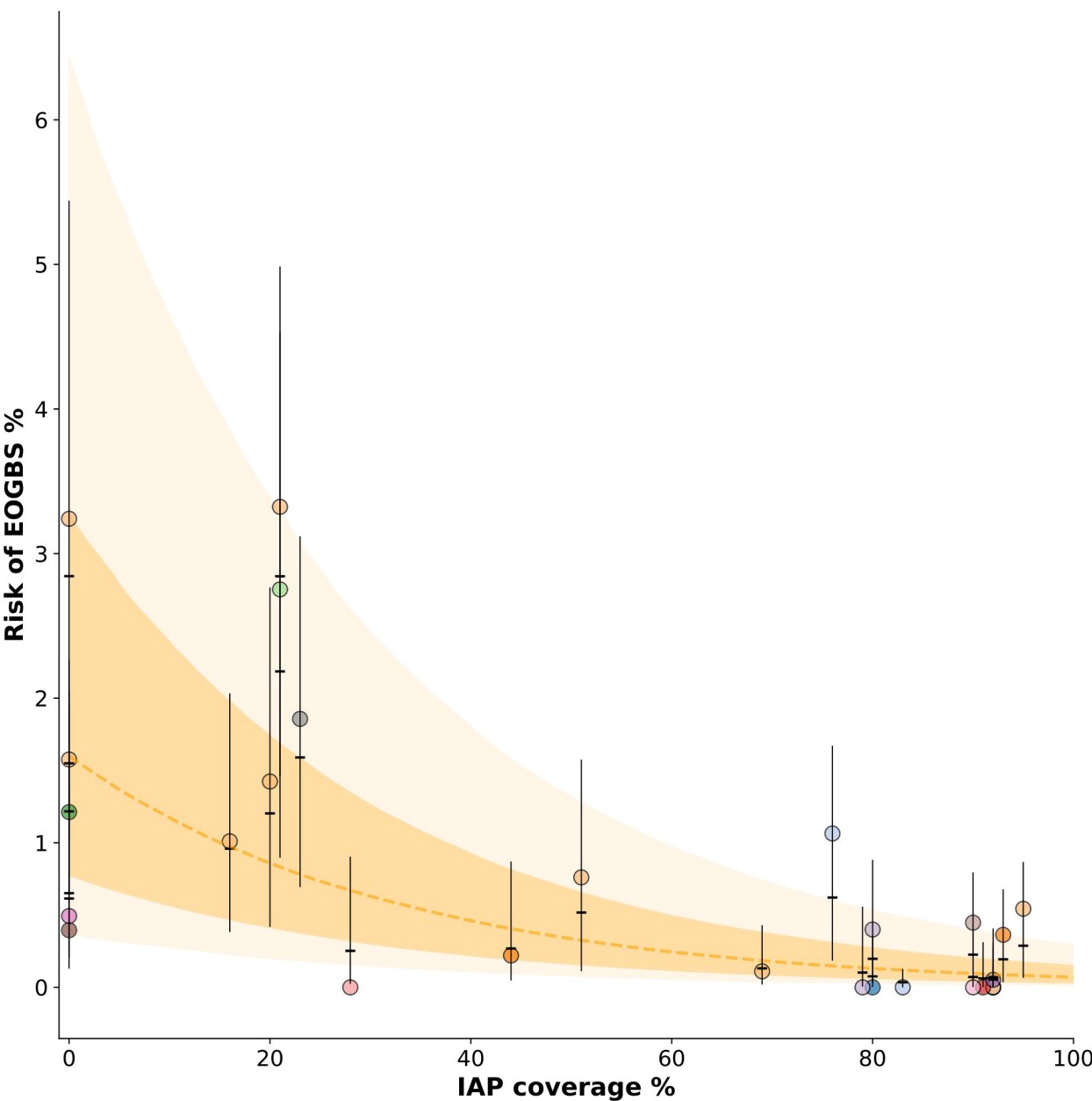

**Fig 4.** Risk of early-onset invasive GBS disease (EOGBS, y-axis) by intrapartum antibiotic coverage (x-axis). For this figure, the model described in the subsection *Early-onset invasive GBS disease in babies born to GBS-colonized mothers* was used. The shaded area represents the 10–90% interval of the risk of early-onset iGBS disease in a new study; the distribution was generated by assuming the intercept for a new study $i$ follows the distribution $\alpha_{i-new}^{(m)} \sim Normal(\alpha_g^{(m)}, \sigma_{eo}^{(m)2})$, which uses estimates of $\alpha_g$ and $\sigma_{eo}$ at each iteration $m$. The darker orange shade represents the interquartile range of the same distribution and the dashed line, the median. Of note, this does not incorporate the component of the predictive probability linked to sampling uncertainty; it only incorporates the component related to estimation uncertainty. Each circle represents a different study included in the analysis, with circles of similar colour corresponding to studies performed in the same country. Posterior medians and 95% intervals of the risk in these studies, i.e. $r_i$ for studies in the analysis dataset, are shown as black lines.

Data from incidence studies were incorporated in the full model I, that combines Eqs 1–6, through the following equations:

$$c_i = p'_{ij} \times invlogit(\alpha'_i + \beta_{IAP}X_{ij}^{IAP}) \times \gamma \qquad (7)$$

$$y_i^{inc} \sim Binomial(c_i, n_i^{inc}) \qquad (8)$$

where $y_i^{inc}$ is the number of early-onset iGBS disease cases observed in the incidence study $i$; $n_i^{inc}$, the total number of births in the study population; $p'_{ij}$ is the predicted prevalence of maternal GBS colonization in a study performed in country $j$, where the logit ($p'_{ij}$) is assumed to be normally distributed with parameters $\mu_j$ and $\sigma_s$; $c_i$ is the estimated study-specific probability of early-onset iGBS disease, based on parameters defined in the previous subsections. Given that incidence studies did not test pregnant women for GBS colonization, an assumption implied in Eq 7 is that prevalence estimates as quantified by sensitive methods, in terms of culture methodology and anatomical sampling site, represent the true underlying maternal GBS prevalence in these study populations. To allow for under-reporting (or under-ascertainment) of cases, we introduced the parameter $\gamma$, that represents the proportion of all cases reported. In the analysis that only included selected studies (full model I), this parameter was assumed to be common for all incidence studies; we used as prior the distribution *Beta ~ (2, 2)* for $\gamma$. In the secondary analysis that included a higher number of incidence studies (full model II), this parameter was allowed to vary (see *Methods* section for details). We used $\alpha'_i$ and $\beta_{IAP}$ to estimate the risk of early-onset iGBS disease in babies born to colonized mothers in the incidence study $i$ population; $\alpha'_i$ follows a Normal distribution with mean $\alpha_g$ and standard deviation $\sigma_{eo}$. $X_{ij}^{IAP}$ represents IAP coverage in the country $j$, assumed to correspond to the coverage in the incidence study $i$ population. This information was obtained from a study by Le Doare and colleagues [20], who reviewed literature and consulted national medical societies to assess IAP policy adoption. In this analysis, we did not differentiate between the prophylaxis approach that involves treatment of pregnant women with evidence of GBS colonization and the approach that uses risk factors to identify mothers who should receive antibiotics.

In the full model I, posterior medians and 95% intervals of the parameters estimated in the subsection *Early-onset invasive GBS disease in babies born to GBS-colonized mothers* changed to -4.17 (-4.69, -3.68) and -0.03 (-0.04, -0.02) for $\alpha_g$ and $\beta_{IAP}$, respectively. Estimates for these same parameters in analyses using full model II were: -4.28 (-4.76, -3.79) and -0.02 (-0.03, -0.017). Changes in coefficients relating country-level covariates and logit maternal GBS colonization prevalence were minor after incorporating incidence data in the model (see **Table A in S1 Appendix**). There were differences in posterior median GBS prevalences when comparing the hierarchical GBS colonization model alone versus full models I and II (**Fig F in S1 Appendix**): as examples, after including data from incidence studies, the posterior median maternal GBS colonization prevalence changed from 23.1 to 23.7% in Australia, 18.7 to 18.1% in Malawi and 20.1 to 20.6% in the United Kingdom. The posterior median (95% interval) of the parameter $\gamma$ was 0.59 (0.34, 0.89) in the analysis that assumed the same parameter for all incidence studies (full model I); using full model II, the following values (posterior medians and 95% intervals) were estimated: 0.51 (0.28, 0.84), 0.38 (0.20, 0.69) and 0.67 (0.38, 0.93) for population-based studies, facility-based studies in developing countries and facility-based studies in developed countries, respectively. For comparison, **Fig 5** shows the numbers of early-onset iGBS disease cases estimated using the different models presented in this manuscript for the 82 countries with maternal GBS colonization data.

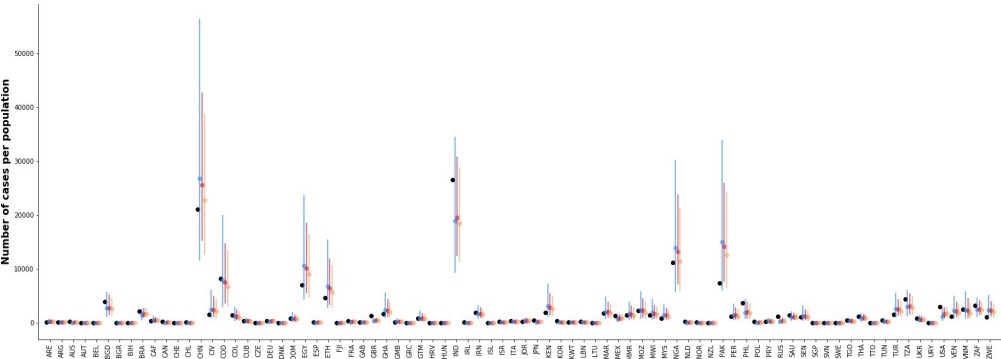

**Fig 5.** Estimated absolute numbers of early-onset invasive GBS disease cases (y-axis). Estimates for the 82 countries (x-axis) with maternal GBS colonization data are shown. Black circles represent the number of cases estimated in the evidence synthesis work by Seale and colleagues; uncertainty ranges reported by the authors are not Bayesian and not included in this figure. Estimates from different models described here correspond to different colours: blue represents Bayesian estimates that do not use incidence data; red represents full model I; and orange, full model II. For comparability, we used the same numbers of live births per country as Seale and colleagues. See also **Fig J in S1 Appendix** for a similar comparison on the incidence scale.

## Discussion

We described a method to estimate country-level incidence of early-onset invasive GBS disease. This method uses studies previously reviewed in [5], where data on maternal GBS colonization were combined with results of epidemiological studies that estimate the risk of early-onset iGBS disease in babies born to colonized mothers. For the estimation of country-level maternal GBS colonization prevalence, we used a hierarchical model, which allowed partial pooling of data from different countries and also allowed to directly account for between-study differences in GBS detection methods. Furthermore, the method described here, unlike the previous review (5), integrated indirect estimates, using data on the proportion of mothers who are GBS colonized, with data from studies that directly estimate incidence of iGBS disease in all live births.

In modelling maternal GBS colonization prevalence, we used a hierarchical model, which has several advantages over standard regression methods [21], including partial pooling of data and inclusion of predictors at different levels. At the study level, our model confirmed and accounted for the association between microbiology diagnostics used in studies and study-level prevalence. As one of the objectives in developing this model was to, in future analyses, estimate GBS colonization prevalence in countries where no studies have been performed, we also included country-level covariates in our model. The following variables were associated with country-level maternal GBS colonization prevalence: HIV prevalence, antibiotics use during lower respiratory infections (LRIs) in children, gross national income per capita, neonatal mortality and obesity prevalence. At the individual level, HIV infection has been linked to GBS infection phenotypes, although a recent meta-analysis suggests that HIV infection does not have a significant effect on maternal GBS carriage [22]. Antibiotics use during LRIs, on the other hand, was included in our analysis as a surrogate for antibiotics availability and use in different countries, which could influence frequencies of bacteria carriage, including GBS. Our results suggest an association between this variable and maternal GBS colonization prevalence at the national level, which should not be interpreted as causal at the individual level [23,24] and could be related to other country-level differences in medical practices if this variable is not a close proxy for antibiotics use during pregnancy. Gross national income per capita is likely associated with other factors that directly influence risk of

colonization and could represent distal determinants of disease as defined by Victora and colleagues [25]. Furthermore, the association between obesity prevalence and country-level maternal GBS colonization prevalence is supported by previous studies that demonstrate an association at the individual level [26]. In future analyses, the posterior distributions of the coefficients for these different variables will be used in the estimation of maternal GBS colonization prevalence for countries without data, which will also incorporate the unexplained between country variation represented by the scale parameter $\sigma_c$. As seen in **Table A in S1 Appendix**, the inclusion of incidence data in our model did not modify country-level regression coefficients.

Studies that assessed the risk of early-onset iGBS disease in neonates born to GBS-colonized mothers showed great variation in the proportion of children developing this medical condition, even when performed in the same country and with the same proportion of the study population receiving antibiotics. This variability is shown in **Fig 4**. As expected, we observed an association with study-level intrapartum antibiotic coverage, with studies with high coverage having an estimated risk below 1%. Pooled individual-level data analyses of these studies might help to understand the heterogeneity in this risk and improve burden estimates.

Epidemiological studies that directly estimate iGBS disease incidence are believed to be biased: for example, studies based on hospital data by design include children attending hospitals with capacity to perform microbiological tests. In low- and middle-income countries, limited access to health care facilities might lead to a non-negligible proportion of cases being missed when passive data capture is used. Another likely mechanism of under-estimation of early-onset iGBS disease incidence relates to the sensitivity of microbiological methods. This might apply to both incidence studies and studies that assess the risk of disease in babies born to GBS-colonized mothers. Indeed, in a study performed in the United Kingdom, the estimated combined incidence of confirmed and presumed early-onset iGBS disease was 3.6 cases in 1,000 births [27], which is 6–7 times higher than the incidence estimated in a recent study using prospective surveillance [9]; and in a review of incidence studies performed in developing countries, Dagnew and colleagues observed that studies with automated culture methods for GBS detection had, on average, higher incidence of iGBS disease compared to studies with manual culture methods [28]. Furthermore, in population-based studies, the comprehensiveness of the reporting from microbiology laboratories might be variable: in England and Wales, increase in reporting was thought to partially explain longitudinal trends in iGBS disease incidence [29]. Our model addresses this issue by integrating iGBS disease incidence data with GBS colonization data and adjusting for under-reporting within an evidence synthesis framework. Including data from incidence studies in our estimation led to changes in parameters of the model, in particular those related to the risk of early-onset iGBS disease in babies born to colonized mothers. Given our model assumptions, we estimated that incidence studies missed, on average, ~40% of cases of iGBS disease that occurred in their study population. This is key in the understanding of the burden of iGBS disease in infants, and there is an urgent need for systematic data collection using different study designs, which can then be compared and integrated within models such as the one we present. This will be crucial to the design of future surveillance strategies for iGBS disease.

In summary, we developed a Bayesian model to estimate the national burden of early-onset iGBS disease that combines different types of epidemiological data available on this important neonatal condition. As shown in **Fig 1**, this allowed epidemiologically relevant parameters to be informed by more than one study type. The efforts to develop a maternal vaccine that could prevent GBS disease in neonates mean that estimations of the total burden of GBS disease are essential to guide clinical development, trial design and policy adoption. Furthermore, these

estimates could be used to inform decisions about IAP policy during a time when there is heightened concern over antibiotic resistance.

## Materials and methods

### Data

**Maternal GBS colonization and early-onset invasive GBS disease risk data.** We used data from two recent (2017) reviews [14,18]. Data on both maternal GBS colonization and culture-confirmed early-onset invasive GBS disease risk in children born to colonized mothers were obtained from the supplementary appendices of these two manuscripts. For data on iGBS disease risk given colonisation, we excluded from this analysis two of the 30 studies described in the review by Russell et al that only included term neonates.

**Data from incidence studies.** In the review by Madrid and colleagues [3], studies with data on the incidence of infant iGBS disease were selected that also reported a population denominator (live births). In that analysis, 14 studies were considered less biased by the authors and were selected for inclusion in our full model I. For two countries with two studies among those considered less biased, we selected the most recent study for inclusion in our analysis. Furthermore, two studies performed in countries without GBS colonization data were not included. In two studies, authors reported corrected numbers of cases after accounting for study design; we used these corrected numbers in our estimation. In a secondary analysis (full model II), we used all incidence studies reported in Madrid et al [3] that included data from 0 to 6 days of life; as above, for countries with more than one incidence study, we included the most recent.

### Maternal GBS colonization model

**Model parameterization.** For the hierarchical model on maternal GBS colonization prevalence, we initially used a centered parameterization, as defined by Betancourt [30]. Since divergences were frequently observed when sampling the posterior distribution (see **Fig H in S1 Appendix** for an example of the distribution of the divergences in the parameter space [31]), we re-parameterized the model using non-centered parameterization.

**Study-level variables.** In this hierarchical model, study-level variables were coded so that high sensitivity methods were coded as 0, and low sensitivity methods, as 1, which allowed a direct estimation of country-level prevalence based on sensitive diagnostics.

**Country-level variables.** Country-level variables were included in this analysis based on the disease process and our knowledge of factors that could influence GBS carriage at the population level. To the list used in Seale et al [5], which includes variables that represent frequencies of individual-level risk factors for GBS colonization, such as obesity and HIV infection, variables that reflect access to care during pregnancy, such as skilled birth attendance coverage and antenatal care coverage, and socio-economic variables (e.g. gross national income per capita, proportion of the population living in urban areas), we added antibiotics use for LRIs in children, as it might represent general antibiotic availability that might influence GBS colonization prevalence. Data were obtained from relevant sources, including the World Health Organization (neonates protected against tetanus, low-birth weight frequency, frequency of cesarean section), the World Bank (gross national income per capita, GINI), the United Nations (urbanization, neonatal mortality rates, fertility rates) and Institute for Health Metrics and Evaluation (frequency of skilled birth attendance, age-standardized obesity prevalence, years of maternal education, antibiotics coverage in lower respiratory infections, antenatal care coverage). Missing values in country-level variables were input using the mean values in each WHO region. **Fig I in S1 Appendix** presents the posterior distributions of the coefficients

in the hierarchical model that included all the country-level variables. The final model included the following variables, for which the 95% posterior interval did not include zero: antibiotics coverage during LRIs, prevalence of female obesity, gross national income per capita and HIV prevalence. In addition, the variables neonatal mortality and maternal education, whose coefficients had absolute median values ~0.2 and that are thought to be associated with health care conditions during the neonatal period and maternal behavior that might influence GBS colonization, respectively, were kept in the final model.

**Additional analyses.**    We performed sensitivity analyses (see **S1** *Appendix* for the list of priors used in sensitivity analyses). Furthermore, in addition to the maternal GBS colonization model presented in the *Results* section, we also fit a model with one additional hierarchical level, WHO region-level. We compared the WAIC of the model presented in the *Results* section and of this model that incorporated region-specific intercept terms in [Eq 1]; since the WAIC values were similar, we decided to present the simpler model, without region-level.

## MCMC algorithm

The models described in the *Results* section were fit to data using Hamiltonian Monte Carlo; results were generated with *PyMC3* and *PyStan* packages in Python; codes are available. Gelman-Rubin diagnostic was used to assess convergence.

## Supporting information

**S1 Appendix. In the S1 Appendix, we described sensitivity analyses and included additional figures and a table with results of model checks and comparisons.** The following tables and figures were included: **Table A.** Maternal GBS colonization regression coefficients at the study and country levels and standard deviation parameters. **Figure A.** Study-specific maternal GBS colonization prevalence (i.e. percentage of study population colonized by GBS bacteria) by diagnostic combinations. **Figure B.** Maternal GBS colonization prevalence model and prior predictive distribution. **Figure C.** Prior predictive distribution of the model on early-onset invasive GBS disease in babies born to GBS-colonized mothers. **Figure D.** Mixed predictive checks of the maternal GBS colonization prevalence model. **Figure E.** Mixed predictive checks of the model on early-onset invasive GBS disease risk. **Figure F.** Posterior median maternal GBS colonization prevalence estimated by the model that only used data from GBS colonization studies and by the full model, that combined these data with early-onset iGBS disease incidence and risk data. **Figure G.** Predictive checks for the full model. **Figure H.** Distribution of divergences in the centered model. **Figure I**. Posterior distributions of regression coefficients in the hierarchical model for maternal GBS colonization that includes all variables. **Figure J.** Estimated country-level incidence of early-onset invasive GBS disease per 1,000 births.
(DOCX)

## Acknowledgments

We would like to thank Anna Seale and Simon Cousens for discussions in early stages of this work.

The views expressed in this publication are those of the authors and not necessarily those of NIHR, PHE, or the UK Department of Health and Social Care.

## Author Contributions

**Conceptualization:** Mark Jit, Joy Lawn.

**Data curation:** Bronner P. Gonçalves, Artemis Koukounari, Proma Paul.

**Formal analysis:** Bronner P. Gonçalves.

**Funding acquisition:** Mark Jit, Joy Lawn.

**Investigation:** Bronner P. Gonçalves, Simon R. Procter, Sam Clifford, Artemis Koukounari, Alexandra Lewin, Mark Jit, Joy Lawn.

**Software:** Bronner P. Gonçalves.

**Writing – original draft:** Bronner P. Gonçalves.

**Writing – review & editing:** Bronner P. Gonçalves, Simon R. Procter, Artemis Koukounari, Proma Paul, Alexandra Lewin, Mark Jit, Joy Lawn.

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
