## [Decision Letter · Decision Letter 0]

17 Feb 2021

Dear Dr. Goncalves,

Thank you very much for submitting your manuscript "Estimation of country-level incidence of early-onset invasive Group B Streptococcus disease in infants using Bayesian methods" for consideration at PLOS Computational Biology.

As with all papers reviewed by the journal, your manuscript was reviewed by members of the editorial board and by several independent reviewers. In light of the reviews (below this email), we would like to invite the resubmission of a significantly-revised version that takes into account the reviewers' comments.

We cannot make any decision about publication until we have seen the revised manuscript and your response to the reviewers' comments. Your revised manuscript is also likely to be sent to reviewers for further evaluation.

Sincerely,

Benjamin Muir Althouse

Associate Editor

PLOS Computational Biology

Nina Fefferman

Deputy Editor

PLOS Computational Biology

Reviewer's Responses to Questions

**Comments to the Authors:**

Reviewer #1: Review was uploaded as an attachment.

Reviewer #2: The authors aimed to estimate the country-level incidence of EOGBS. The have also updated published methods using a hierarchical Bayesian approach. First, the statistical methods seem very appropriate for the task and I commend the authors the thought and effort that has gone into this work. It will represent an important contribution to the field of newborn health.

•The use of antibiotic coverage for children with respiratory infections seems to be a poor proxy for antibiotic use among pregnant women (lines 185-186). I understand the limitations in relevant data and the need to account for antibiotic use. However, the authors could discuss the implications, including directionality, associated with using this proxy.

•Related to antibiotic use, Figure 2 seems to indicate that median colonization prevalence from the literature is lowest in largely low- and middle-income countries. Am I reading this correctly? I would assume this has to do with exposure to antibiotics before being sampled in such countries. But I would be interested to hear the author’s perspectives on this. Would this need to be accounted for?

•Last, would it be possible to extend these findings to other countries? Could the authors comment on what might be needed extend this model to cover additional countries?

Minor points

•On line 57, the authors should provide a citation for Group B Streptococcus being a major cause of morbidity or mortality. Or tone down this language.

•On lines 60-61, the authors state that “only few studies had tried to assess…” But only one reference is given. Reference 5 also corresponds with a multi-country analysis and should be provided here.

•Beginning on line 65, it would help the clarity to note who used knowledge on the pathogenesis. Was this Seale, et al?

•For figure 5, could the authors consider a figure that captures incidence as well? Most countries seem to be right at 0 EOGBS cases, but I wonder how this would change if it were per 1000 live births.

**Have all data underlying the figures and results presented in the manuscript been provided?**

Reviewer #1: Yes

Reviewer #2: Yes

PLOS authors have the option to publish the peer review history of their article (what does this mean?). If published, this will include your full peer review and any attached files.

Reviewer #1: No

Reviewer #2: No
---

## [Decision Letter · Decision Letter 1]

25 Apr 2021

Dear Dr. Goncalves,

We are pleased to inform you that your manuscript 'Estimation of country-level incidence of early-onset invasive Group B Streptococcus disease in infants using Bayesian methods' has been provisionally accepted for publication in PLOS Computational Biology.

Best regards,

Benjamin Muir Althouse

Associate Editor

PLOS Computational Biology

Nina Fefferman

Deputy Editor

PLOS Computational Biology

Reviewer's Responses to Questions

**Comments to the Authors:**

Reviewer #1: The authors have appropriately responded to all reviewer concerns.

Reviewer #2: I am satisfied with the responses to my questions/comments.

**Have the authors made all data and (if applicable) computational code underlying the findings in their manuscript fully available?**

Reviewer #1: Yes

Reviewer #2: Yes

PLOS authors have the option to publish the peer review history of their article (what does this mean?). If published, this will include your full peer review and any attached files.

Reviewer #1: No

Reviewer #2: No

---

## [Editor Report · Acceptance letter]

28 May 2021

PCOMPBIOL-D-20-02028R1 

Estimation of country-level incidence of early-onset invasive Group B Streptococcus disease in infants using Bayesian methods

Dear Dr Gonçalves,

I am pleased to inform you that your manuscript has been formally accepted for publication in PLOS Computational Biology. Your manuscript is now with our production department and you will be notified of the publication date in due course.

With kind regards,

Kata Acsay
